# A Caveat When Using Alkyl Halides as Tagging Agents to Detect/Quantify Reactive Sulfur Species

**DOI:** 10.3390/antiox11081583

**Published:** 2022-08-16

**Authors:** Xiaohua Wu, Yuping Xin, Qingda Wang, Yongzhen Xia, Luying Xun, Huaiwei Liu

**Affiliations:** 1State Key Laboratory of Microbial Technology, Shandong University, Qingdao 266200, China; 2School of Molecular Biosciences, Washington State University, Pullman, WA 991647520, USA

**Keywords:** RSS, alkyl halides, mBBr, IAB

## Abstract

Using alkyl halides to tag reactive sulfur species (RSSs) (H_2_S, per/polysulfide, and protein-SSH) is an extensively applied approach. The underlying supposition is that, as with thiols, RSS reacts with alkyl halides via a nucleophilic substitution reaction. We found that this supposition is facing a challenge. RSS also initiates a reductive dehalogenation reaction, which generates the reduced unloaded tag and oxidized RSS. Therefore, RSS content in bio-samples might be underestimated, and its species might not be precisely determined when using alkyl halide agents for its analysis. To calculate to the extent of this underestimation, further studies are still required.

## 1. Introduction

In the recent two decades, a new class of sulfur-containing compounds termed reactive sulfur species (RSS) have been discovered in mammalian cells, serum, and other tissue liquids, as well as plants and microorganisms [1]. Hydrogen sulfide (H_2_S), hydrogen per/polysulfide (HS_n_H, *n* ≥ 2), and glutathione per/polysulide (GS_n_H, *n* ≥ 2) are three of the most abundant RSSs detected in mammals [2]. Among them, H_2_S has been deemed as the third gaseous signaling molecule after CO and NO [3,4]. HS_n_H and GS_n_H directly modify cysteine residues to form sulfhydrated proteins (protein-SSH) [5]. This modification has been observed in key regulators including protein 1 (Sp1), Caspase 3, and Kelch-like ECH-associated protein 1 (Keap1) [6]. Furthermore, sulfhydration modification of these proteins is related to important physiological processes. For instance, SP1 transcriptional activity is enhanced upon sulfhydration at Cys 68 and Cys 755 residues, leading to the upregulation of vascular endothelial growth factor receptor 2 and neuropilin-1, and finally affecting vascular function and health [7]. Sulfhydration of caspase-3 inhibits its activity and thereby suppresses cancer cell apoptosis [8]. Sulfhydration of Keap1 at Cys151 residue significantly suppresses vascular atherosclerosis accelerated by diabetes [9].

In eukaryotic cells, the metabolism of RSS is compartmentalized with cytoplasm and mitochondria, which are considered to be the two major places. A few RSS-metabolizing enzymes have been identified [10]. Cystathionine beta-synthase (Cbs) and cystathionine gamma-lyase (Cse) mainly locate incytoplasm; 3-mercaptopyruvate sulfurtransferase (3-Mst) and cysteinyl-tRNA synthetase 2 (Crs2) mainly locate in mitochondria. These four enzymes use cysteine or its oxidized form, cystine, as a precursor to generate RSS. Differently, sulfide:quinone oxidoreductase (Sqr) is located in the mitochondrial inner membrane, which catalyzes the oxidation of hydrogen sulfide to per/polysulfide. Cellular excessive per/polysulfide is oxidized by persulfide dioxygenase (Pdo) to sulfite. This enzyme is located in the mitochondrial matrix in mammalian cells. Recently, it has been observed that impairing RSS metabolism by knocking down Crs2 leads to obvious mitochondrial dysfunction, indicating that RSS metabolism is highly related with mitochondrial health [11].

"RSS exerts various beneficial effects including cytoprotection, anti-inflammation, vasodilation, angiogenesis, and cardioprotection [5,12]. Currently, the biggest challenge in studying its functions is the development of reliable methods for its detection and quantification [13]. The use of alkyl halides to tag thiols such as GSH has a long history [14]. The underlying mechanism is thiols’ ability to initiate nucleophilic attack toward the halogen-linked C atom, resulting in the formation of a C–S bond and left of the halogen anion (Figure 1A, reaction 1). When developing RSS detection/quantification methods, this strategy was adopted [13,15]. The fluorescent probe monobromobimane (mBBr) is now extensively used to tag H_2_S and per/polysulfide based on the supposition that, as thiols, RSS can replace the bromine atom via a nucleophilic substitution reaction (Figure 1A, reactions 2–6). The product RSS-mB can be separated by reversed-phase chromatography and then detected by a fluorescence detector. For protein-SSH assay, the alkyl iodide probes, including iodoacetyl-PEG2-biotin (IAB) and iodo-N-(prop-2-yn-1-yl) acetamide (IPM), are widely used. These probes tag both protein-SH and protein-SSH (Figure 1A, reactions 7,8). The protein-SS tag can be enriched and analyzed via mass spectrometry. A few proteome-wide protein-SSH analysis methods have been developed using these probes [16,17].

Although such alkyl halide-based methods are now seen in almost all of the RSS-related literature their reliability has not been thoroughly investigated. Previously, we observed that, when using mBBr to tag RSS, unknown peaks constantly showed in LC chromatograph. In addition, when we used IAB to tag and enrich protein-SSH, >90% of proteins obtained were mis-tagged [18]. These observations suggest that the reactions between alkyl halides and RSSs are more complicated than between alkyl halides and thiols. Accompanied with the advances in RSS chemistry study, we now know that RSS is more reductive than thiols [19]. Therefore, a question is emerging: is RSS active enough to reduce alkyl halides? i.e., can RSS remove the halide atom from the probes via a reductive dehalogenation reaction (Figure 1B)? Since the reduction reaction generates a reduced tag (Figure 1C,D) other than the RSS tag derivative, a second question follows: if it happens, to what degree it will impair the reliability of the alkyl halide-based RSS detection/quantification methods? In this study, we tried to address these questions.

## 2. Materials and Methods

### 2.1. Chemicals

Sodium hydrosulfide (NaHS), GSH, GSSG, and mBBr were purchased from Sigma–Aldrich (Shanghai, China). IAB was purchased from Thermo Fisher (Shanghai, China). Other reagents including octasulfur (S_8_), NaCl, NaOH, and EDTA were all purchased from Sigma–Aldrich (Shanghai, China). HSSH was prepared using a reported method [20]. Briefly, 25.6 mg S_8_ powder, 44.8 mg NaHS, and 32 mg NaOH were mixed in 20 mL deionized water. The solution was blown with argon gas for 15 min to remove oxygen and then placed in a 37 °C incubator for 48 h under anaerobic condition. The prepared HSSH were quantified by the cyanide method [21]. GSSH was prepared following a reported method [22]. Briefly, 10 mM GSSG was mixed with a 50 mM NaHS in KPi buffer (10 mM, pH 7.0, with 200 mM NaCl and 1 mM EDTA) under anaerobic condition. The reaction was performed at 30 °C for 30 min. The prepared GSSH was quantified by the cyanide method.

### 2.2. Conditions of the Reactions

For GSH + mBBr reaction, two molar ratios were used. The 1:1 ratio was performed with 50 μL GSH (8 mM) and 50 μL mBBr (8 mM), which were mixed in distilled water. The 1:10 ratio was performed with 50 μL GSH (2.5 mM) and 50 μL mBBr (25 mM) in distilled water. The reactions were carried out in a 1.5 mL-scale brown centrifuge tube to avoid light photolysis. After incubating the mixture at room temperature for 30 min, 50 μL acetonitrile was added to terminate the reaction. The mixture was then centrifuged at 12,000 rpm for 3 min. The supernatant was analyzed by high-performance liquid chromatography coupled with tandem mass spectrometry (HPLC-MS/MS).

For GSSH + mBBr reaction, two molar ratios were used. The 1:1 ratio was performed with 50 μL GSSH (8 mM) and 50 μL mBBr (8 mM). The 1:10 ratio was performed with 50 μL GSSH (2.5 mM) and 50 μL mBBr (25 mM). Other conditions were the same as for GSH + mBBr reaction.

For H_2_S + mBBr reaction, two molar ratios were used. The 1:1 ratio was performed with 50 μL NaHS (8 mM) and 50 μL mBBr (8 mM). The 1:10 ratio was performed with 50 μL NaHS (2.5 mM) and 50 μL mBBr (25 mM). Other conditions were the same as for GSH + mBBr reaction.

For HSSH + mBBr reaction, two molar ratios were used. The 1:1 ratio was performed with 50 μL HSSH (8 mM) and 50 μL mBBr (8 mM). The 1:10 ratio was performed with 50 μL HSSH (2.5 mM) and 50 μL mBBr (25 mM). Other conditions were the same as for GSH + mBBr reaction.

For GSH + IAB reaction, 50 μL GSH (8 mM) and 50 μL IAB (8 mM) were mixed in Tris-HCl buffer (50 mM, pH 8.3, with 5 mM EDTA). Other conditions were the same as for GSH + mBBr reaction.

For GSSH + IAB reaction, two molar ratios were used. The 1:1 ratio was performed with 50 μL GSSH (8 mM) and 50 μL IAB (8 mM). The 1:10 ratio was performed with 50 μL GSSH (8 mM) and 50 μL IAB (80 mM). Other conditions were the same as for GSH + IAB reaction.

For S_8_ + mBBr reaction, we firstly dissolved S_8_ in methanol to make a 17 mM solution; then, 10 μL S_8_ (17 mM) and 90 μL mBBr (1.9 mM) were mixed in distilled water. Other conditions were the same as for GSH + mBBr reaction. For S_8_ + IAB reaction, 10 μL S_8_ (17 mM) and 90 μL IAB (1.9 mM) were mixed in Tris-HCl buffer (50 mM, pH 8.3, with 5 mM EDTA). Other conditions were the same as for GSH + mBBr reaction.

### 2.3. HPLC-MS/MS Analysis

The samples were analyzed using a 126-quadrupole time-of-flight, high-resolution mass spectrometer (Ultimate 3000, Burker impact HD, Thermo Fisher, Waltham, MA, USA). The column used was InertSustain C18 5 μm (4.6 mm × 250 mm, GL Sciences, Shanghai, China). The injection volume was 10 μL. The ESI mass spectrometer (Thermo Fisher, Waltham, MA, USA) was used with the source temperature set at 200 °C and the ion spray voltage at 4.5 kv. Nitrogen was used as the nebulizer and drying gas. The mobile phase was a mixture of pure water and methanol. The column flow rate was set at 0.8 mL/min and the column temperature was maintained at 38 °C. The liquid elution procedure was as below (Table 1).

## 3. Results

We first performed thiol/RSS + mBBr reactions and carefully checked their products. The molar ratio of RSS to mBBr was 1:1 or 1:10. HPLC-MS/MS measurements were carried out to analyze the products. Chemical structures of the products were shown in Figure 2. Both calculated and detected molecular weights (MWs) were given. The MS^1^ signal intensities were used for a preliminary quantification of them. The first reaction was GSH + mBBr under 1:1 condition. As expected, GS-mB (tagged product) was the dominating product. Although mBH (reduced mBBr) was also detected, its MS^1^ signal intensity was very low. The ratio of GS-mB to mBH was 100:0.8 according to their MS^1^ signal intensities (Table 2). Under 1:10 condition, the ratio of GS-mB to mBH was 100:0.2. These results indicated that GSH cannot efficiently reduce mBBr, and that it mainly participates in the nucleophilic substitution reaction. The foundation of alkyl halide-based, thiol-tagging methods stood the test.

For RSS + mBBr reactions, the first one was GSSH. The tagged product GSS-mB was still the major product; however, the portion of reduced product mBH significantly increased (Table 2). The ratios of GSS-mB to mBH were 100:5.2 (under 1:1 condition) and 100:15.4 (under 1:10 condition). H_2_S was the second one. The ratios of tagged product (mB-S-mB) to mBH were 100:24.9 and 100:23.9, under 1:1 and 1:10 conditions respectively. The third one was HSSH. The ratios of the tagged product (mB-SS-mB) to mBH were 100:208.5 and 100:78.7, under 1:1 and 1:10 conditions respectively.

Second, we performed RSS + IAB reactions. The products were also analyzed via HPLC-MS/MS. Chemical structures of the products were shown in Figure 3. Both calculated and detected MWs were given. The MS^1^ signal intensities were used for a preliminary quantification. For GSH, the ratio of GS-AB (tagged product) to HAB (reduced product) was 100:3.8; however, for GSSH, the ratios of GSS-AB (tagged product) to HAB were 100:92.9 and 100:83.4, under 1:1 and 1:10 conditions, respectively (Table 2). GSSH generated much more HAB than GSH.

Cyclic octasulfur (S_8_) is the ultimate oxidized form of HS_n_H. We performed S_8_ + mBBr and S_8_ + IAB reactions under both 1:1 and 1:10 conditions; however, only trace amounts of mBH or HAB were detected (MS^1^ signal intensity < 10^2^), indicating that RSS without the H atom cannot reduce alkyl halides.

Considering MBH and HAB are representing products of the reductive dehalogenation reaction, we checked their MS^2^ signals. The detected fragments were shown in Figure 4. Although no authentic standards for reference are available. The distribution patterns of daughter ions verified that these two products were correct. Combining all analysis, we confirmed that RSS indeed can initiate reductive dehalogenation reactions when reacting with alkyl halide agents.

## 4. Discussion

In this work, we studied the reactions between RSS and alkyl halides. Among the three reactive sulfur species tested herein, HSSH led to more reduced product (mBH) than GSSH and H_2_S. Compared to RSS, GSH showed relatively weak reluctivity and led to a very small ratio of reduced product. The substitutive product GS-mB or GS-AB was much more abundant, indicating that substitutive reactions are dominating when GSH meets mBBr or IAB, and that the reductive reactions can be neglected when using these tags to detect/quantify GSH. However, in the case of RSS, reductive reactions matter significantly, and therefore cannot be neglected. The ratio of reduced products can even be higher than substituted ones. It is noteworthy that, herein, we used MS^1^ signal intensities to represent the products’ relative amounts due to a lack of authentic standards. Although it is not an accurate quantification method, it still can reflect their relative abundance because these chemicals share common structural frameworks.

The reductive reactions can also generate oxidized products. Theoretically, for H_2_S involved reactions, the oxidized product should be HSSH, and, for HSSH, the oxidized product should be HS_4_H. However, HS_n_H is super unstable in aqueous solution. Polymerization and depolymerization reactions occur rapidly. It has been suggested that using an alkylating probe to examine the chain length distribution of RSS is under the Curtin–Hammett control, meaning that the probe breaks the internal dynamic equilibrium, and the detected distribution does not really reflect the situation in the solution [23]. Therefore, we did not examine oxidized RSS product and believed that using the content of oxidized RSS to assess the reduction reaction was not proper. The content of the reduced agent (mBH and HAB) was a more precise marker when estimating how much RSS was consumed in the reduction reaction.

Previous to our study, a few reports mentioned that RSS can initiate dehalogenation reactions when encountered with halogenated alkyls. For instance, Bondarenko et al. discovered that polysulfides (HS_n_H, *n* = 2~5) rapidly react with methyl iodide (MeI), 1,3-dichloropropene (1,3-D), and chloropicrin (CP). Analysis of reaction kinetics and initial products suggests that the reaction is SN2 nucleophilic substitution for MeI and 1,3-D but likely reductive dehalogenation for CP [24]. Zhang et al. studied the reactions between H_2_S/HS_n_H and hexabromocyclododecane (HBCDD) [25]. They observed that both RSS chemicals can release bromine atoms from HBCDD via reductive debromination. They also reported that the reaction of HBCDD with HS_n_H is about six times faster than with H_2_S, which is consistent with our observations. However, they did not mention the substitutive product, which is different from our results. The reason may lie in the fact that they used a very high HS_n_H/RSS ratio (~30/1), which resulted in mainly reductive products. These reports were not noticed by RSS researchers before, probably because they are published in journals of the environmental engineering field. On the other hand, these reports suggest that RSS may have various applications in other fields due to their specific activities.

Bianco et al. studied the activity of RSSH and examined the ability of RSSH species to act as one-electron reductants [26]. They found that RSSH is relatively easily oxidized, compared to thiols, by weak oxidants to generate the perthiyl radical (RSS·). Surprisingly, RSS· was found to be stable in the presence of both O_2_ and NO and only appears to dimerize. Thus, the RSSH/RSS· can be an active redox couple. Chauvin et al. also reported that RSSH are much more reactive to two-electron oxidants than thiols [27]. They investigated the H-atom transfer chemistry of RSSH, contrasting it with the well-known H-atom transfer chemistry of thiols, and found that RSSH are excellent H-atom donors, besting thiols by as little as 1 order and as much as 4 orders of magnitude. In the same report, the authors suggested that the inherently high reactivity of RSSH to H-atom transfer is based largely on thermodynamic factors, the weak RSS−H bond dissociation enthalpy (~70 kcal/mol), and the associated high stability of the perthiyl radical. Based on these studies, we proposed that RSSH can participate in both single and double electron transfer reactions. At a low-pH condition (pH < p*K*a), RSSH mainly presents in protonated form (RSSH); in this case, it reacts with alkyl halides via H-atom transfer mechanism (signal electron transfer). Whereas, when pH is higher than its p*K*a, RSSH mainly presents in deprotonated form (RSS^-^). The nucleophilic RSS^-^ attacks alkyl halides and reacts with them via double electron transfer. Therefore, the solution pH condition may affect the reaction mechanisms. However, no matter which mechanism, both can lead to reductive products (Figure 5). It is noteworthy that, for detecting RSS in biological samples, alkyl halides are often added in excess amount. We performed the reactions with two ratios. However, both 1:1 and 1:10 ratios led to reductive products. How the ratio affects the distribution of products requires further investigation.

## 5. Conclusions

In conclusion, our experiments undoubtedly proved that, compared with thiols, RSS has significantly higher reductive activity toward alkylating probes. This property makes RSS initiate the reductive dehalogenation reduction. Therefore, the same foundation cannot be simply transplanted to the alkyl halide-based RSS-tagging methods. The neglect of the reduction reaction causes at least two issues: first, the amounts of RSS in bio-samples become underestimated. Second, most oxidized, long chain-RSS may be generated during the alkylation step instead of actually presenting in bio-samples.

## Figures and Tables

**Figure 1 antioxidants-11-01583-f001:**
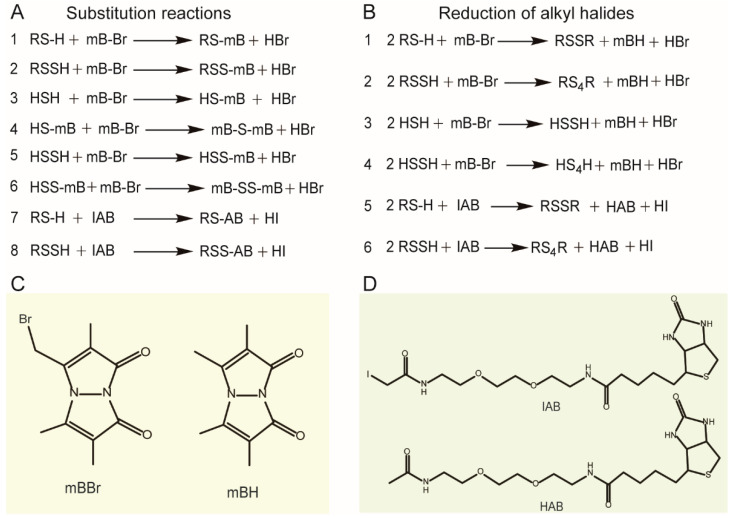
The substitution reactions between alkyl halide probes and RSS (**A**) and reduction in alkyl halides by RSS (**B**). The structures of alkyl halide probes and their reduced products are shown (**C**,**D**).

**Figure 2 antioxidants-11-01583-f002:**
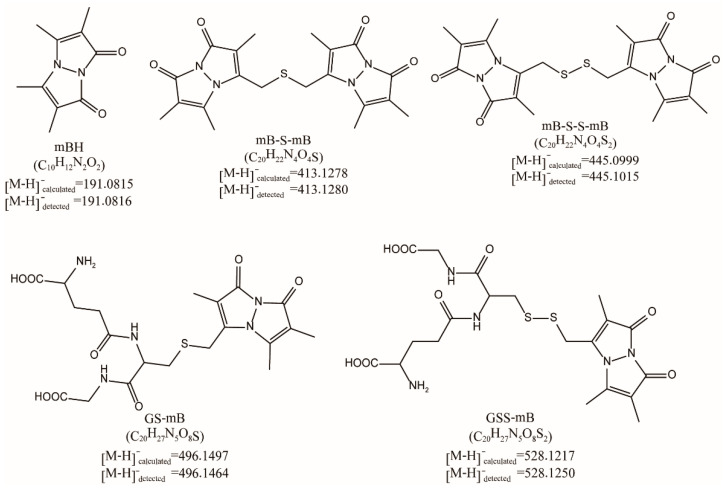
Structure, calculated MWs, and detected MWs of the mBBr derived products.

**Figure 3 antioxidants-11-01583-f003:**
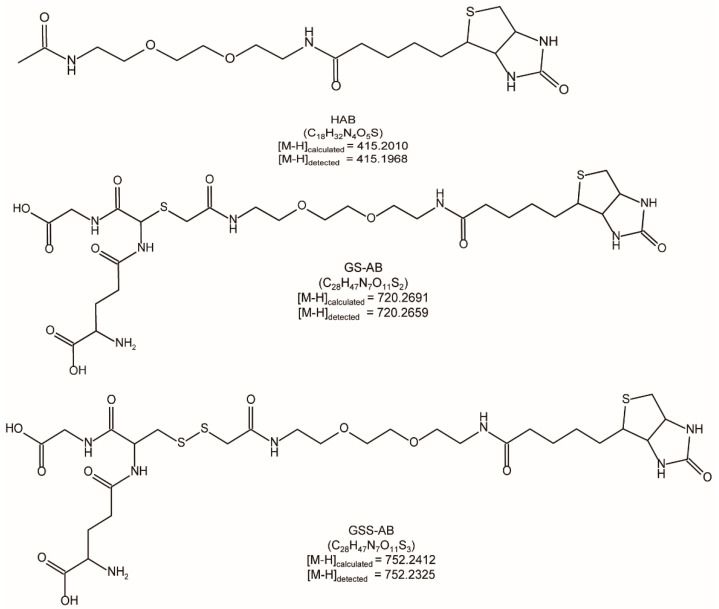
Structure, calculated MWs, and detected MWs of the IAB derived products.

**Figure 4 antioxidants-11-01583-f004:**
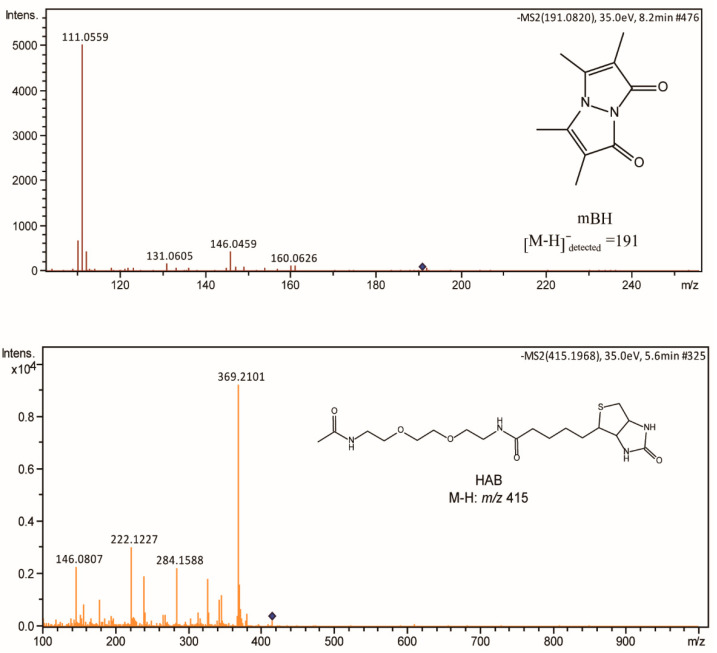
MS^2^ spectra of mBH and HAB detected in this study.

**Figure 5 antioxidants-11-01583-f005:**
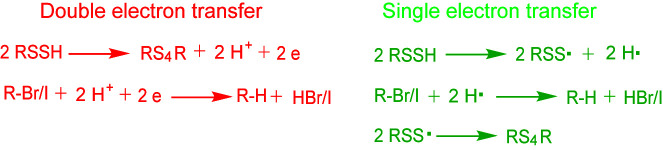
Proposed reaction mechanisms between RSSH and alkyl halides.

**Table 1 antioxidants-11-01583-t001:** Liquid elution procedure of HPLC-MS/MS analysis.

Time (min)	Pure Water (%)	Methanol (%)
0.01	92.5	7.5
1	47.5	52.5
15	45	55
15.1	0	100
20	0	100
20.1	92.5	7.5
31	92.5	7.5

The ESI mass spectrometer was used with the source temperature set at 200 °C and the ion spray voltage at 4.5 kv. Nitrogen was used as the nebulizer and drying gas.

**Table 2 antioxidants-11-01583-t002:** MS^1^ signal intensities of tagged and reduced products.

Reaction	Tagged Product	Reduced Product
GSH + mBBr (1:1)	GS-mB: 8.97 × 10^5^	mBH: 7.77 × 10^3^
GSH + mBBr (1:10)	GS-mB: 3.06 × 10^5^	mBH: 8.52 × 10^2^
GSSH+ mBBr (1:1)	GSS-mB: 4.59 × 10^6^	mBH: 2.37 × 10^5^
GSSH + mBBr (1:10)	GSS-mB: 2.88 × 10^6^	mBH: 4.45 × 10^5^
H_2_S + mBBr (1:1)	mB-S-mB: 1.59 × 10^5^	mBH: 3.97 × 10^4^
H_2_S + mBBr (1:10)	mB-S-mB: 1.66 × 10^5^	mBH: 3.98 × 10^4^
HSSH + mBBr (1:1)	mB-SS-mB: 4.70 × 10^4^	mBH: 9.80 × 10^4^
HSSH + mBBr (1:10)	mB-SS-mB: 4.83 × 10^4^	mBH: 3.80 × 10^4^
GSH + IAB (1:1)	GS-AB: 7.89 × 10^5^	HAB: 3.01 × 10^4^
GSSH + IAB (1:1)	GSS-AB: 3.52 × 10^6^	HAB: 3.27 × 10^6^
GSSH + IAB (1:10)	GSS-AB: 2.30 × 10^6^	HAB: 1.92 × 10^6^

## Data Availability

All data mentioned in this study are available from corresponding authors upon reasonable requirements.

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
