# Peer review of "A Caveat When Using Alkyl Halides as Tagging Agents to Detect/Quantify Reactive Sulfur Species"

_antioxidants, 2022, doi:10.3390/antiox11081583_

Round 1
Reviewer 1 Report
The paper by Wu et al. submitted to the Antioxidants is entitled: A caveat when using alkyl halides as tagging agents to detect/quantify reactive sulfur species. Based on the results of their study, the Authors suggest that reactive sulfur species, including GSSH or HSSH exhibit reducing properties leading to reduction of alkyl halides besides substitution reaction. It is important information due to the fact that alkyl halides, like mBBr, are commonly used for quantitative assay of thiols and per/polysulfides.
In this context, the paper by Wu et al. is valuable and innovative, however, there are some aspects that the Authors should consider and address.
Introduction
I wonder if protein-SSH is needed as a keyword considering that the Authors in the reported study used non-protein RSS including HSSH, GSSH and H2S. On the other hand, IAB should be added to the keywords.
In the introduction (line 24-25), the information about the importance of S-sulfhydration should be extended. Providing more details on the proteins regulated by persulfidation will be valuable. Sp1 or Keap1 should be introduced and their role should be discussed taking into account their persulfidation.
Figure 1 – The reaction equations in part B are inconsistent (2 thiols or RSS are needed to reduce one mBBr). Parts C and D should be corrected to equal size. I suggest that part B should be entitled “Reduction of alkyl halides” (from the point of view of thiols or RSS, this is an oxidation reaction).
Results
-Why wasn’t the ratio 1:10 tested in the case of GSH and HSSH?
- The Table 2 – spaces in ratio 1:1 or 1:10 are too large.
-The paragraph concerning S8 does not match the continuity of the reported results. It should appear at the end of the Results section. Why weren’t the results of experiments with S8 included in the Table 2?
-Figure 2 and 3 should be of the similar style.
Discussion
Generally, discussion, like the whole manuscript is short and concise and it can be expanded a lot. The Authors should discuss their results comparing them with other reports (e.g. Bianco et al. examined the ability of RSSH to act as one-electron reductants and compared them with corresponding thiols: Free Radic Biol Med 2016, 101: 20-31).
Line 148 – the Authors wrote that HSSH is more reductive than GSSH and H2S and cited the references 5 and 17, however, I have not found this information in the cited references. It should be reconsidered.
Are there some factors, like pH or others that can influence the RSS reactivity? It could be tested or at least hypothetically discussed. Moreover, in the case of some RSS (like GSSH with mBBr) the ratio of RSS and alkyl halide (1:1 and 1:10) influences the ratio of tagged and reduced product, while in the case of other RSS, it does not (i.e. H2S with mBBr and GSSH with IAB). Do you have any idea to explain this fact?
Minor points:
There are some grammar and typing errors – the manuscript needs general correction, i.e.:
Line 26 – RSS are reactive sulfur species (plural), so “exerts” should be replaced by “exert”
Line 33 - monobromobimane, not monobormobiane
Line 106 – the word “alkyl” is there redundant
The voltage unit should be expressed by a capital letter (kV instead of kv).
According the Antioxidants instructions for Authors, in the text, reference numbers should be placed in square brackets [ ] before the punctuation, not as superscripts after punctuation. It should be corrected throughout the manuscript.
Reviewer 2 Report
This is a really interesting subject and provided with sufficient quality to be published in the "Antioxidants" journal. A very brief Introduction contains the information necessary to follow the urgency of the studied problem. The authors properly selected references and limited them to last years.
The manuscript content matches the scope of the journal although, in my opinion, it is rather a letter/communicate/short report on the observed inconsistence of the analytical method of quantitative detection of reactive sulfur species, and if it was the letter, the current degree and depth of description would be enough (assuming that letter will be followed by full length paper with extended information on mechanisms, the detailed calculation, including statistical analysis, of possible error produced by this methodology etc. During reading the results and discussion sections I could not recognize whether the authors intend to create an analytical paper (and propose how to solve the problem, with calculated errors the current methodology produces) or just to inform us that they discovered a gap in the currently used analytical methodology, but they leave this problem for their future studies or for other researchers to solve the problem.
Anyway, the subject is interesting and worth to be published.
minor comments:
line 143-146: please add a phrase with example(s) of reductive dehalogenation observed in similar systems/compounds and add reference – even from organic chemistry texbooks or reviews, that would support your hypothesis about dehalogenation (the best way would be to insert a possible mechanism of dehalogenation here).
line 100: replace the colon by full stop, because Table 1 cannot be included as a part of the phrase.
Reviewer 3 Report
In this manuscript, Liu and coworkers wish to report an important observation of reactive sulfur species (RSS)-mediated reduction of alkyl-halide probes to generate reduced unloaded probes. Since alkyl halides have been extensively used for RSS measurement, the results presented here show the limitations of alkyl halides for estimating the accurate concentration of RSS in biological samples and caution must be taken. The analytical experiments were thoroughly carried out. Overall, this is an interesting study and merits publication in antioxidants after addressing the comments below.
1. Please indicate the time of GSSG reaction with H2S to produce the GSSH. The GSSH generation method used here involves several equilibrium reactions and the presence of additional products such as H2S, GSH, and glutathione trisulfide. In addition to GSSH, can other sulfur species reduce alkyl halides?
2. “RSS exerts various beneficial effects including cytoprotection, anti-inflammation, vasodilation, and angiogenesis.” In this sentence, the authors may also add 'cardioprotection'. In this sentence, references are missing. Please cite appropriate references.
3. The ratio of reduced products over tagged products was high for H2S and HSSH reactions with mBBr. Why the authors did not try H2S and HSSH reactions with IAB?
4. The MS1 signal intensities were used for a preliminary quantification of products. If the ionization efficiency of these compounds is not the same, the ratio of the products reported here may not be accurate. Can the authors comment on this?
5. “We performed S8 + mBBr and S8 + IAB reactions under both 1:1 and 1:10 conditions, but only trace amount of mBH or HAB were detected (MS1 signal intensity <102), indicating that RSS without the H atom cannot reduce alkyl halides.” Which solvent was used for this reaction? If it is water, does S8 soluble in this reaction medium?
6. The reviewer is not suggesting additional experiments, but the authors may include the oxidized RSS information from the same HPLC MS/MS data.
7. Please define Sp1 and Keap1. Typo “monobormobiane”
Round 2
Reviewer 1 Report
In the present, revised form this manuscript can be accepted for publication.